# Both Amyloid-β Peptide and Tau Protein Are Affected by an Anti-Amyloid-β Antibody Fragment in Elderly 3xTg-AD Mice

**DOI:** 10.3390/ijms21186630

**Published:** 2020-09-10

**Authors:** Alejandro R. Roda, Laia Montoliu-Gaya, Gabriel Serra-Mir, Sandra Villegas

**Affiliations:** 1Protein Design and Immunotherapy Group, Departament de Bioquímica i Biologia Molecular, Facultat de Biociències, Universitat Autònoma de Barcelona, Bellaterra, 08193 Barcelona, Spain; alejandro.ramos@uab.cat (A.R.R.); laia.montoliu.gaya@gu.se (L.M.-G.); gabriel.serra@uab.cat (G.S.-M.); 2Department of Psychiatry and Neurochemistry, Institute of Neuroscience and Physiology, The Sahlgrenska Academy at the University of Gothenburg, 431 41 Mölndal, Sweden

**Keywords:** Aβ, tau, immunotherapy, Alzheimer, scFv, 3xTg-AD

## Abstract

Alzheimer’s disease (AD) is the most common dementia worldwide. According to the amyloid hypothesis, the early accumulation of the Aβ-peptide triggers tau phosphorylation, synaptic dysfunction, and eventually neuronal death leading to cognitive impairment, as well as behavioral and psychological symptoms of dementia. ScFv-h3D6 is a single-chain variable fragment that has already shown its ability to diminish the amyloid burden in 5-month-old 3xTg-AD mice. However, tau pathology is not evident at this early stage of the disease in this mouse model. In this study, the effects of scFv-h3D6 on Aβ and tau pathologies have been assessed in 22-month-old 3xTg-AD mice. Briefly, 3xTg-AD female mice were treated for 2 weeks with scFv-h3D6 and compared with 3xTg-AD and non-transgenic (NTg) mice treated with PBS. The treatment with scFv-h3D6 was unequivocally effective in reducing the area of Aβ staining. Furthermore, a tendency for a reduction in tau levels was also observed after treatment that points to the interplay between Aβ and tau pathologies. The pro-inflammatory state observed in the 3xTg-AD mice did not progress after scFv-h3D6 treatment. In addition, the treatment did not alter the levels of apolipoprotein E or apolipoprotein J. Thus, a 2-week treatment with scFv-h3D6 was able to reduce AD-like pathology in elderly 3xTg-AD female mice.

## 1. Introduction

Alzheimer’s disease (AD) is the most common cause of dementia worldwide. Alzheimer’s International (ADI) estimated that more than 50 million people are suffering dementia worldwide nowadays and that the global economic cost of dementia for the year 2018 was above US$ 1 trillion [1]. Incidence rates of any dementia and AD are reported to be greater in women than men [2]. As an example, the incidence rate of AD in Europe was 11.08 per 1000 person-years; 13.25 per 1000 person-years in women, and 7.02 per 1000 person-years in men [3]. In consonance, the prevalence of AD was 7.13% in women and 3.31% in men (overall, 5.05%). Because the prevalence of AD is estimated to be tripled in 30 years, the finding of disease-modifying drugs is mandatory.

AD is a chronic and progressive neuropsychiatric disorder characterized by memory loss, cognitive impairment, and behavioral and psychological symptoms of dementia, including anxiety, phobia, and irritability, owing to synaptic loss and neuronal death. Although its etiology has not been completely unraveled yet, the main histopathological hallmarks of AD include the extracellular accumulation of Aβ-peptide in senile plaques and the formation of neurofibrillary tangles (NFTs) of hyperphosphorylated tau protein [4].

According to the amyloid hypothesis, the early accumulation of Aβ in the brain triggers the pathogenic cascade leading to tau hyperphosphorylation, neuroinflammation, synaptic loss, and neuronal death [5], although other factors could also be involved [6]. Indeed, it is not clear yet whether Aβ is a cause or a consequence of the pathological process. Even though the strong evidence of the role of Aβ in familial AD supports the amyloid hypothesis, drugs targeting Aβ or presenilins have not been successful. Therefore, to design more accurate drugs, it is mandatory to continue investigating more deeply the role of Aβ [7]. Although amyloid plaques and NFTs were previously widely believed to be related to neurodegeneration, accumulating evidence indicates that the oligomeric species of Aβ and tau are the most neurotoxic species [8]. Intracellular Aβ deposition precedes extracellular Aβ accumulation and tau hyperphosphorylation both in human AD brains and in transgenic mice [9,10]. Thus, enhancing Aβ clearance has become one of the main approaches in pharmacological research intended to cease or slow down the progression of AD [11]. Several anti-Aβ monoclonal antibodies (mAbs) have been tested in clinical trials, bapineuzumab being the first one to reach phase III [12]. Although Aβ immunotherapy was effective in animal models [13], some adverse effects appeared during clinical trials. The occurrence of amyloid-related imaging abnormalities (ARIAs) owing to edema was evident in apolipoprotein E4 (*apoE4*) carriers receiving the highest doses of bapineuzumab, together with the absence of detectable clinical benefit, which led to the discontinuation of clinical trials [14]. Because these side effects are likely related to an immune over-activation via the Fc-receptor, the use of single-chain variable fragments (scFv) that lack the Fc fragment has emerged as an alternative to immunization with full-length mAbs [15].

The anti-Aβ single-chain variable fragment scFv-h3D6, in which the V_H_ and V_L_ domains are fused by a (Gly_4_Ser)_3_ peptide linker, is derived from bapineuzumab [16,17], thus, targeting the five N-terminal residues of Aβ and detecting all the different aggregation states (monomers, oligomers, and fibrils). ScFv-h3D6 prevented Aβ-induced toxicity in SH-SY5Y neuroblastoma cell cultures by withdrawing Aβ oligomers from the amyloid pathway and redirecting them to a different, non-toxic, worm-like pathway [16,18,19]. Moreover, we have widely demonstrated that the early intervention with scFv-h3D6 to 5-month-old 3xTg-AD female mice ameliorates the first hallmarks of AD, without eliciting any detectable inflammatory response. A single intraperitoneal (i.p.) administration protected against neuronal death in the deep cerebellar nuclei [20], reduced the extra- and intracellular Aβ burden, and restored apoE and apoJ levels [21] in the 3xTg-AD mouse brain. More importantly, such a single administration prevented the neuronal loss in the hippocampus and amygdala and improved spatial memory, even though behavioral and psychological symptoms of dementia (BPSD)-like symptoms were not ameliorated, while neither astrogliosis nor microgliosis was induced [18,22]. Apart from being safe in terms of neuroinflammation, the treatment did not affect kidney or liver function [22].

Likewise, magnetic resonance imaging revealed the tendency of the scFv-h3D6 treatment to protect against further reduction in brain volume than already existing in 5-month-old 3xTg-AD mice [23]. This observation was corroborated by a recent longitudinal study, from 5 to 12 months of age, where we found a striking fact. Treatment with scFv-h3D6 by some means affected the endogenous Aβ levels also in the non-transgenic mice. Although the significance found in the 3xTg-AD group was not reached, we observed a tendency for the protection from the loss of volume associated with aging in the cortex, cerebrum, and the whole brain, and diminished the 6E10+ intensity, amyloid load, and IL-6 values in both the hippocampus and, especially, the cortex [24]. Indeed, in our first study in vivo [21], the treatment was able to improve the performance of the non-transgenic animals in some learning tasks, although the overall effect on behavior/cognition was much less evident than in the 3xTg-AD mice.

It is noteworthy that scFv-h3D6 crossed the blood–brain barrier and was internalized by the Aβ peptide-containing neurons in the early phase post-injection while being co-localized with the Aβ-peptide in glial cells in the late phase post-injection [25]. These findings showed that the mechanism underlying the Aβ-peptide withdrawal is the engulfment of the scFv-h3D6/Aβ complex by the glial cells.

Because AD is usually diagnosed after extracellular Aβ aggregates and tau pathologies become evident, the assessment of the therapeutic potential of any putative drug at the advanced stages of pathology is of capital relevance, particularly concerning the ability of an anti-Aβ antibody to ameliorate tau pathology. In this sense, tau immunotherapy has been shown to reduce amyloid deposition [26,27], which supports the hypothesis of the occurrence of a cross-talk between Aβ and tau. Thus, the searching for pharmacological approaches targeting both histopathological markers is mandatory. The aim of the present work is to evaluate the efficacy of scFv-h3D6 treatment in older animals with evident senile plaques and NFTs. For 2 weeks, 22-month-old 3xTg-AD female mice were treated with either scFv-h3D6 or PBS and, then, Aβ and tau levels were quantified, along with several other markers related to neuroinflammation.

## 2. Results

### 2.1. ScFv-h3D6 Accumulates in the Hippocampus

Previous studies have shown that anti-Aβ antibodies were mainly accumulated in the central periventricular areas, especially in the hippocampus of AD mouse models, where Aβ load is high [28]. To assess whether scFv-h3D6 is homogenously distributed or accumulated in the hippocampus, immunohistostaining of brain sections from treated 3xTg-AD mice was performed. ScFv-h3D6 was accumulated in the hippocampus of the treated mice (Figure 1A), whereas a very faint labeling was detected in the cortex (Figure 1B). This very faint labeling was negligible and corresponds to a residual cross-reactivity of the polyclonal antibody used, as observed by us previously [22], because the same intensity of labeling was observed in samples from untreated animals (Figure 1C). Moreover, scFv-h3D6 co-localized with Aβ, not only in the extracellular plaques of the hippocampus (tM1 = 0.342, max = 0.545, min = 0.192; tM2 = 0.187, max = 0.326, min = 0.098) but also, and more intensely, inside the *cornus amonis* 3 (CA3) neurons (tM1 = 0.597, max = 0.781, min = 0.349; tM2 = 0.476, max = 0.511, min = 0.382) (Figure 1A, arrows).

### 2.2. ScFv-h3D6 Reduces the Area of 6E10 Staining

We have previously reported that scFv-h3D6 reduces extracellular Aβ oligomers [21] and intracellular Aβ levels [22] after a single intraperitoneal dose to 5-month-old 3xTg-AD mice. We could also observe these effects after treatment for 6 weeks to a similar group of animals [23]. However, the effect of the scFv-h3D6 in elderly animals with an advanced stage of AD-like pathology, where plaques are present, has not yet been evaluated.

Many senile plaques were observed in the hippocampus of 22-month-old 3xTg-AD mice, especially in the subiculum (Figure 2A, arrows). After scFv-h3D6 treatment, the percentage of 6E10-immunoreactive area (Figure 2B) was significantly reduced (*U*-value = 0; *p* = 0.0079; r = 1). However, in spite of the median for the number of Aβ extracellular aggregates was lower after treatment (Figure 2C), significance was not reached (*U*-value = 5; *p* = 0.1508; r = 0.6). Quantification of Aβ_40_ (Figure 2D–F) and Aβ_42_ (Figure 2G–I) levels in the different hippocampal protein extracts by ELISA showed that the levels of both Aβ_40_ and Aβ_42_ were negligible in NTg mice. Even though both Aβ_40_ and Aβ_42_ levels were reduced after scFv-h3D6 immunization to 3xTg-AD mice in all the protein fractions (TBS-soluble, SDS-soluble, and FA-soluble), differences were not significant, except for Aβ_40_ in the SDS-soluble fraction, where the difference was marginally significant, and effect size indicated this difference is real (*U*-value = 3; *p* = 0.056; r = 0.76). It is also worth noting that the Aβ load was higher in the SDS- and FA-soluble fractions than in the TBS-soluble fraction, as expected at this advanced stage of pathology.

### 2.3. ScFv-h3D6 Decreases Total Tau Levels

Because 3xTg-AD mice develop both histological hallmarks of AD, senile plaques composed of Aβ, and NFT of tau protein [29], and considering that bapineuzumab was able to reduce tau levels in the CSF from patients with mild-to-moderate AD [30], the effect of scFv-h3D6 in tau pathology was evaluated.

Localization of Aβ and total tau in the hippocampus was examined (Figure 3). Aβ was detected extra- and intracellularly whereas NTFs were located intracellularly within Aβ-containing neurons in the CA1 of the hippocampus (tM2 = 0.932, max = 0.982, min = 0.893). Since amyloid pathology precedes NFTs formation, not all 6E10 reactive neurons were NFTs positive (tM1 = 0.087, max = 0.101, min = 0.079).

When focusing on tau and the effect of scFv-h3D6, the 3xTg-AD, but not NTg mice, showed strong labeling for mAb HT7 (*U*-value = 0; *p* = 0.0079; r = 1), indicative of high levels of total tau, which were partially decreased after treatment (*U*-value = 4; *p* = 0.0952; r = 0.68) (Figure 4A,B). When using mAb AT8 directed against Ser202 and Thr205 phosphorylated tau, the labeling before the treatment was not strong and the effect of treatment was not so evident (Figure 4C). Phosphorylated tau was also increased in 3xTg-AD compared to NTg mice (*U*-value = 0; *p* = 0.0079; r = 1)) but no effect of scFv-h3D6 treatment was observed (*U*-value = 10; *p* = 0.8016; r = 0.12) (Figure 4D). Therefore, total and phosphorylated tau levels in the SDS-soluble fraction of the hippocampus were determined by WB (Figure 4E–G). Total tau levels were negligible in the NTg and elevated in the untreated 3xTg-AD group (*U*-value = 0; *p* = 0.0079; r = 1) (Figure 4E,G). After the treatment, a tendency for a reduction in the total tau levels was found and the effect size showed these reductions as real (*U*-value = 4; *p* = 0.0952; r = 0.68). However, non-significant differences in phosphorylated tau levels, as quantified with the mAb AT8, were observed after the treatment (*U*-value = 8; *p* = 0.5317; r = 0.28) (Figure 4F,G). It should be noted that a large range in tau levels is inherent to the 3xTg-AD model for both total and phosphorylated tau and that scFv-h3D6 treatment reduced this dispersion (Figure 4E–G). This means that treatment with an anti-Aβ scFv reduced overall tau load.

### 2.4. ScFv-h3D6 Slightly Ameliorates the Pro-Inflammatory Status

Several studies support the role of the immune system in neurodegeneration, as reviewed by reference [31]. There are some proteins involved in the immunological processes, such as the triggering receptor expressed on myeloid cells 2 (TREM2), that are associated with an increased risk of AD onset [32]. Besides, the administration of full-length antibodies promotes astroglia and microglia reactivity and an overall inflammatory process [33]. Considering the importance of the immune system and the propensity of its over-activation after Aβ-immunotherapy, the effect of scFv-h3D6 treatment on inflammation in 22-month-old 3xTg-AD female mice has been studied here.

In Figure 5, co-localization of different combinations of astroglia or microglia and Aβ peptide or tau protein in the hippocampus of 3xTg-AD mice are shown. In the first row (Figure 5A), reactive astrocytes (green) are found surrounding Aβ plaques (red). This concurs with wrapping the soma of some Aβ-containing neurons by astrocytes and microglia reported in a previous deep study of neuronal populations in the 3xTg-AD model [10]. In the second row (Figure 5B), amoeboid microglia (green) are found close to Aβ plaques (red). Similarly, in the third (Figure 5C) and fourth (Figure 5D) rows, astroglia and microglia, respectively, surround tau positive neurons in the CA1. Thus, the interplay among Aβ, tau, and glial cells becomes apparent.

GFAP and Iba1 levels, markers for astroglia and microglia, respectively, were measured by WB from the TBS-soluble fraction (Figure 6A–C). GFAP levels were increased in 3xTg-AD with respect to NTg mice (Figure 6A,C), although the difference was only marginally significant (*U*-value = 3; *p* = 0.0556; r = 0.76). Interestingly, after scFv-h3D6 treatment, GFAP levels tended to decrease to that of the NTg mice (*U*-value = 4; *p* = 0.0952; r = 0.68). Such a difference was also observed by immunohistostaining (Figure 6D,E), where untreated 3xTg-AD mice exhibited strong GFAP labeling (*U*-value = 3; *p* = 0.0556; r = 0.76) that was decreased after treatment (*U*-value = 2; *p* = 0.0317; r = 0.84). Concerning Iba1 levels, no differences were detected by immunohistochemistry between NTg and 3xTg-AD mice (*U*-value = 9; *p* = 0.7857; r = 0.12 (Figure 6F,G) or WB (*U*-value = 5; *p* = 0.1508; r = 0.6) (Figure 6B,C). Iba1 levels detected by immunohistochemistry (*U*-value = 6; *p* = 0.2222; r = 0.52) or WB (*U*-value = 8; *p* = 0.51317; r = 0.28) were not increased after scFv-h3D6 administration.

To further analyze the inflammatory state of treated and untreated 3xTg-AD mice, the levels of TNFα, IL-33, IL-1β, and IL-6 were measured in the TBS-soluble fraction (Figure 6H–K). TNFα levels tended to be increased in the untreated 3xTg-AD group, and effect size revealed that the difference is real (*U*-value = 4; *p* = 0.0952; r = 0.68), but not in the treated 3xTg-AD animals (Figure 6H). IL-33 was also increased in untreated 3xTg-AD mice (*U*-value = 1; *p* = 0.0159; r = 0.92) with respect to NTg mice, and no differences were found between treated and untreated 3xTg-AD mice (Figure 6I). However, a tendency to be equal was found between treated 3xTg-AD and NTg, and effect size indicated they are actually equal (*U*-value = 4; *p* = 0.0952; r = 0.68, not in Table 1). Similarly, IL-1β was elevated in untreated 3xTg-AD mice (*U*-value = 0; *p* = 0.0079; r = 1), and in treated 3xTg-AD mice, there were no effects, although it should be noted that there was an outlier point masking putative differences (Figure 6J). Finally, no differences in IL-6 levels were found among the groups (Figure 6K).

### 2.5. Effect of ScFv-h3D6 on Apolipoproteins’ Levels

Increasing evidence supports the role of apoE, apoJ, and their common receptor LRP1, in promoting Aβ clearance [34]. Considering this, ELISA was used to assess whether the scFv-h3D6 treatment modulates apolipoproteins levels and WB was used to check the effect of the treatment on LRP1 expression. Differences among experimental groups were not detected for apoJ levels or for the LRP1 expression (Figure 7B,C). However, increased levels of apoE were detected (Figure 7A) in 3xTg-AD mice in comparison with NTg mice (*U*-value = 2; *p* = 0.0317; r = 0.84). Although apoE reduction was previously reported after scFv-h3D6 treatment in 5-month-old 3xTg-AD mice [21], there was a small decrease in the median value at 22 months of age that never tended to be close to the values of the NTg group (*U*-value = 10; *p* = 0.6667; r = 0.2).

## 3. Discussion

The knowledge about proteins and pathways involved in AD has exponentially grown in the last few years. However, the translation of therapeutic strategies from preclinical studies to clinical assays has not been successful [35]. The reason for this lack of positive results in clinical trials is not well understood, but several factors that could influence the outcome have been discussed. Among them, targeting Aβ or tau pathologies, instead of targeting both simultaneously, and the advanced stage of the disease when the treatment started during the trials, could explain this apparent standstill.

Bapineuzumab, the humanized form of the murine mAb 3D6, directed to the N-terminal Aβ region, is able to bind both fibrillar and soluble Aβ species [36]. Even though a reduction in Aβ and tau levels was achieved after bapineuzumab administration, severe side effects, together with the lack of positive outcomes for cognition, forced the discontinuation of the trials [14]. Therefore, in order to avoid the adverse effects associated with Fc-dependent microglia over-activation, scFv-h3D6 lacking the Fc region was constructed [16]. A single administration to 5-month-old 3xTg-AD female mice, when pathology was in an early stage, was able to reduce the first hallmarks of AD, and improve the cognitive impairment associated with an overload of intraneuronal Aβ [21,22]. ScFv-h3D6 also decreased Aβ uptake by the human primary astrocytes that should be beneficial because sustained Aβ uptake by astrocytes may impair their normal functions [37,38]. However, nowadays, AD is usually diagnosed when Aβ burden is high, and symptomatology is evident [4]. Thus, studying the effect of scFv-h3D6 administration in elderly animals, where Aβ and tau pathologies develop, is of capital relevance for assessing the therapeutic potential of this antibody fragment.

At this initial point in the discussion, an important methodological issue is worth mentioning. As we [18,21,22,23,24,25] and other authors [39,40,41,42] have extensively argued, although 6E10 mAb is able to recognize Aβ, β-CTD, and APP, what we are measuring in the 3xTg-AD mouse with this mAb is nowadays agreed to be the Aβ peptide. The present work constitutes another demonstration of this fact since scFv-h3D6, which exclusively recognizes the Aβ peptide, promotes a comparable decrease in both 6E10-labeling and Aβ40- and Aβ42-specific signals by ELISA.

Concerning mAbs pharmacokinetics, it has been reported that peripherally administered antibodies appear in the CSF [30], which would indicate that the mAbs may enter into the brain by crossing the blood-CSF barrier, rather than the blood–brain–barrier (BBB) [43]. If this were true, scFvs levels would be higher in the central periventricular areas and almost non-detectable in the cortex. ScFv-h3D6 labeling was detected in the hippocampus of treated 3xTg-AD but not in the cortex, supporting this hypothesis. Therefore, the use of bispecific antibodies combining an anti-Aβ mAb with an anti-transferrin R antibody that has been used as a BBB transporter, which could improve the distribution of therapeutic mAbs in the brain, spreading its activity and increasing the efficacy of treatment [28]. As recently reported, APP can be relocated into transferrin-positive recycling endosomes. Thus, the use of anti-transferrin R antibodies is able not only to promote transport through the BBB but also to drive the therapy directly to an APP-containing organelle [44].

Since scFv-h3D6 was mainly found in the hippocampus, we would refer hereinafter to this brain region. After a relatively short period of treatment, a reduction in Aβ burden was achieved as quantified by immunohistochemistry. Although the median value for the number of senile plaques was reduced in the 3xTg-AD brains after scFv-h3D6 treatment, there was not a significant difference. Importantly, the total 6E10-immunoreactive area was significantly decreased after treatment. Therefore, scFv-h3D6 clears amyloid aggregates not only from the intracellular space [22] but also from the extracellular space. This is in agreement with the reduction in Aβ_40_ oligomers, i.e., trimers, hexamers, nonamers, and dodecamers, previously reported in 5-month-old 3xTg-AD mice [21]. In this work, Aβ_40_ and Aβ_42_ levels were also reduced in TBS-soluble, SDS-soluble, and FA-soluble fractions and, although the differences were not significant, the trend observed for the decrease in Aβ_40_ in the SDS-fraction was revealed as real after considering the effect size. In addition, the trend observed in all the fractions concurs with immunohistochemical analyses, which indicate that scFv-h3D6 reduces Aβ burden, even at this late stage of pathology. It could be possible that anti-Aβ antibodies were retained in senile plaques [28], hindering the removal of oligomers. This could also partially explain why Aβ immunotherapy is not as effective as expected; if antibodies were trapped in Aβ fibrils, oligomers, the most toxic species, could not be targeted. Thus, a higher doses, an increased time-period of treatment, or a redesign of the antibodies to avoid fibril binding and enhance removal of oligomers, could improve the efficacy of the treatment.

Importantly, not only Aβ but mainly hyperphosphorylated tau protein correlates with cognitive impairment associated with AD progression [45,46]. Thus, reducing tau levels is critical for ameliorating dementia-related symptoms. The interplay between Aβ and tau is well documented [47]. On the one hand, Aβ and APP intracellular domain (AICD) can promote tau hyperphosphorylation [48]. On the other hand, some pathogenic characteristics of transgenic models are rescued by ablating the tau gene [49]. Considering that bapineuzumab was effective in removing tau protein from the CSF [30], we investigated whether scFv-h3D6 can reduce tau burden in the hippocampus. A tendency to reduce total tau levels was achieved after scFv-h3D6 treatment (*p* ≤ 0.1), and effect size indicated this reduction is real (r ≥ 0.68). It is important to note that 2 weeks of treatment was effective at a very advanced stage of the disease when tau pathology was well established. Even though a reduction in Ser202 and Thr205 phosphorylated tau was not observed after scFv-h3D6 treatment, future studies analyzing other phospho-tau epitopes could shed light on the effects of anti-Aβ immunotherapy on tau pathology. In a recent study, active full-length DNA Aβ_42_ immunization to 3xTg-AD mice was able to reduce tau levels [50]. However, there are two main differences with our study; 1) they treated animals for 20 months whereas we did so for 2 weeks and 2) our rout of administration (i.p.) was safer than active DNA immunization. How anti-Aβ antibodies could influence the clearance of tau is not well understood, but it is likely a consequence of Aβ reduction. Because both pathologies are linked, it is not surprising that reducing one of them could result in the amelioration of the other [51].

Several mechanisms for explaining how anti-Aβ antibodies reduce amyloid pathology have been proposed. The peripheral sink hypothesis proposes that circulating mAbs could reduce Aβ from blood and other tissues, and thus promote its efflux from the brain to the blood. In support of the peripheral sink hypothesis, Aβ_40_ levels in blood were increased after bapineuzumab administration [52]. Although this mechanism could be shared between bapineuzumab and the scFv-h3D6 derived fragment, it has been demonstrated that the antibody fragment could enter the brain and localize in both the extracellular and intracellular compartments. Importantly, the lack of the Fc domain prevents glial activation via Fc receptors and consequently would avoid the increased inflammatory response observed with full-length mAbs [53].

Inflammatory response and neurodegeneration are linked in AD [54]. The exact role that the immune system plays in triggering AD and progression is not completely understood. However, it is well known that the immune system plays a key role in AD pathogenesis. It has been reported that both complement and microglia mediate synaptic loss [55], that some variants of TREM2 increase the risk for developing AD [56], and that there are astroglial and microglial reactive cells surrounding Aβ plaques [57]. Moreover, this pro-inflammatory state is enhanced after some anti-Aβ treatments as a consequence of glia over-activation [58,59].

In this work, we report the effect of scFv-h3D6 on the inflammatory state of 22-month-old 3xTg-AD female mice brains. Astroglia and microglia were found surrounding senile plaques and NFTs in the hippocampus, as expected. Moreover, an increased GFAP, but not an Iba1 signal, was detected in 3xTg-AD mice, which indicates the occurrence of astrogliosis in this mouse model, but not microgliosis. IL-1β and TNF-α, both secreted by astrocytes [60], were increased in old 3xTg-AD mice. IL-33 was also increased in this model. IL-33 administration to APP/PS1 double-transgenic mouse model of AD could ameliorate cognitive impairment and reduce both Aβ_40_ and Aβ_42_ in the cortex [61]. Besides, it has been reported that IL-33 ablation causes tau accumulation and late-onset neurodegeneration in the cortex and hippocampus [62]. These data suggest that IL-33 is protective against AD. Importantly, elevated expression of IL-33 has been found in the vicinity of Aβ plaques and tau NFTs [63]. Thus, the increased levels of IL-33 in the hippocampus of 22-month-old 3xTg-AD female mice could be a natural response to counteract the progression of the disease. After repeated scFv-h3D6 injections, interleukin levels were not altered with respect to those in the untreated 3xTg-AD. Neither Iba1 nor GFAP were increased, which means that no over-activation of the immune system after scFv-h3D6 administration occurred. Moreover, levels of GFAP tended to decrease (*p* ≤ 0.1) after treatment, and effect size pointed to a real decrease (r ≥ 0.68), and although differences in interleukin levels were not significant, their values were closer to the NTg interleukin levels than to the untreated 3xTg-AD animals’ levels.

Finally, considering that apoE and apoJ are involved in AD pathology and that we previously found apolipoprotein levels increased in 5-month-old 3xTg-AD female mice [21], the same study has been performed at 22-months of age. In this case, only apoE levels were found elevated in the 3xTg-AD group, and no effect of the treatment was observed. In fact, it is interesting to note that the role of apoE in AD is not well understood, and it has been suggested that apoE could be detrimental at early stages and neuroprotective when the pathology is advanced [64]. Thus, in response to oxidative stress, aging, brain damage, or Aβ deposition, neurons synthesize increasing amounts of apoE, which, in turn, undergoes proteolytic processing, generating fragments that cause mitochondrial dysfunction, NFTs formation, and eventually neurodegeneration [65,66].

Therefore, scFv-h3D6 presents many advantages in comparison with the full-length mAb from which it derives such a greater penetrance into the CNS [22] and no over-activation of the immune system [18,22], even at the late stage of the disease, as demonstrated in this work. Thus, the use of antibody fragments could be a safe strategy to be translated to the treatment of other neurodegenerative diseases. However, the fact that Aβ-immunotherapy has not proved an effect in phase 3 could be related to the additional involvement of other non-amyloidogenic or -taugenic mechanisms in AD pathophysiology. Therefore, combined therapies against the amyloid peptide and/or tau with immune system modulators and/or other alternatives such as apolipoprotein mimetic peptides [23,37,38], could constitute the solution for an effective AD treatment.

As a general conclusion, scFv-h3D6 ameliorates the histological hallmarks of AD in 22-month-old 3xTg-AD female mice, without eliciting any detectable inflammatory response. It is important to remark that although other antibodies improve performance in behavioral testing, they do not affect overall amyloid plaque and tau burden in 22-month-old 3xTg-AD [67]. Future studies, targeting both Aβ and tau pathologies at the same time or in combination with other approaches, would improve the efficacy of Aβ-immunotherapy and could constitute an improved pharmacological approach.

## 4. Materials and Methods

### 4.1. ScFv-h3D6 Production

ScFv-h3D6 was produced as previously described [18]. Briefly, the *scFv-h3D6* gene was cloned into the pPicZαA vector (Invitrogen, Carlsbad, CA, USA) and expressed in KM71H *P. pastoris* cells. Cells were grown until an OD_600_ of 2.0 was reached. Then, methanol was supplemented every 24 h to induce protein expression. After 48 h of expression, cells were centrifuged at 3000× *g* and 4 °C for 10 min. ScFv-h3D6 was precipitated with ammonium sulfate. After centrifugation at 100,000× *g* and 4 °C for 1 h, the pellet was re-suspended in 10 mM Na_2_HPO_4_ (pH 6.5) and double dialyzed. Finally, cationic exchange chromatography was performed in a Resource S column coupled to a UP10 AKTA (GE Healthcare, Chicago, IL, USA). Pure scFv-h3D6 was dialyzed to PBS (pH 7.4) and stored at −20 °C until further use.

### 4.2. Animals

The triple-transgenic mouse model of AD (3xTg-AD), harboring *PS1_M146V_*, *APP_Swe_,* and *tau_P301L_* transgenes, was initially engineered at the University of California, Irvine [29]. Both the 3xTg-AD colony and their corresponding non-transgenic (NTg) littermates with the same genetic background (B6129SF2) were purchased from the Jackson Laboratory (Bar Harbour, ME, USA) and bred at the Animal Facility of our university (Autonomous University of Barcelona, UAB, Barcelona, Spain). Female mice were selected because, in contrast to male mice, they conserve the phenotype as originally described [68] and exhibit an exacerbated AD-like pathology [69]. In addition, AD incidence was reported to be greater in women than men [2]. 3xTg-AD mice progressively displayed Aβ deposition, with intraneuronal immunoreactivity at 3-month-old and extracellular deposition at 6-month-old, and tau aggregation at 12-month-old. Cognitive impairment was first detected at 4-month-old as long-term memory impairment [29,70]. No physical alterations were detected in this mouse model with a mean lifespan of 386.9 ± 161.2 days for male mice and 517.1 ± 197.1 days for female mice [71]. Five animals (with the same genotype and sex) were maintained in Makrolon home cages 35 cm × 35 cm × 25 cm at a temperature of 22 ± 2 °C, with a relative humidity of 55 ± 5%, 12 h light:dark cycle starting at 08:00 am, and food and water ad libitum.

### 4.3. Experimental Design and Statistical Analysis

Ten 22-month-old 3xTg-AD female mice were randomly distributed into 2 experimental groups (*n* = 5): Untreated triple-transgenic mice (3xTg/−) and treated triple-transgenic mice (3xTg/+). Simultaneously, the untreated non-transgenic mice (NTg/−) group constituted of 5 age- and gender-matched NTg mice. A treated non-transgenic mice (NTg/+) group was not included because of 3 reasons derived from observations explained in the introduction Section. First, we already demonstrated that the treatment to NTg mice was safe (as well as to 3xTg-AD mice). Second, the benefit observed in the NTg/+ groups in different works never reached significance, and thus no effect was expected in older animals. Third, we focused on the interplay between Aβ and tau pathologies in this work, and the NTg group did not develop tau pathology.

Animals were intraperitoneally administered every 3 days with 200 µg of scFv-h3D6 (~6.6 mg/kg) diluted in 200 µL of vehicle (PBS, pH 7.4), or with 200 µL of vehicle (referred as untreated for the sake of clarity), for 15 days. This was in contrast to previous studies, where a single dose of 3.3 mg/kg was administered to 5-month-old mice [18,20,21,22,25] or repeatedly for 6 weeks since mice were 4.5 month-old [23].

All the experiments were performed in accordance with the requirements of the Committee on Ethics in Animal and Human Experimentation at UAB (CEEAH 0661). All experiments were approved by the UAB Animal Research Committee and the Government of Catalonia on May 15th 2020.

The degree of overlapping between Aβ and scFv-h3D6 signals in co-localization experiments was calculated using Mander’s coefficient [72] from 3 sections per animal. tM1 means the proportion of 6E10 signal that overlaps with anti-scFv-h3D6 signal, and tM2 means the proportion of anti-scFv-h3D6 signal that overlaps with the 6E10 signal. Similarly, the degree of overlapping between Aβ and tau was shown as tM1, meaning the proportion of 6E10 signal that overlaps with the HT7 signal, and tM2, meaning the proportion of HT7 signal that overlaps with the 6E10 signal. Values were expressed as medians, maximum, and minimum. Mander’s coefficient = 1 implied total overlapping, and Mander’s coefficient = 0 implied no overlapping.

For the rest of the experiments, statistical differences among groups were evaluated with the non-parametric Kruskal-Wallis test. Next, a two-to-two comparison was performed using the Mann-Whitney *U*-test. Values were expressed by medians, maximum, and minimum, and depicted as boxplots. A *p*-value ≤ 0.05 was considered statistically significant. A *p*-value between 0.051 and 0.100 was considered a tendency [73]. The effect size was measured by computing Rank-biserial correlation coefficients (Table 1) with the use of the Wendt formula: r = 1 − (2U/n_1_n_2_); where *U* is the Mann-Whitney -statistics, and *n*_1_ and *n*_2_ the sizes of the samples being compared. The correlation r expresses the difference between the proportions of pairs that support the hypothesis, minus the proportion of pairs that do not. For *n*_1_ = 5 and *n*_2_ = 5 the minimum value of r significant at *p*-value ≤ 0.05 is 0.68 [74]. It is important to note that retaining alive 10 22-month-old 3xTg-AD female mice born the same week was rather challenging since this mice model begins to die at 15-month-old.

Graphical and statistical analyses were performed using Graphpad Prism v6 (GraphPad Software, San Diego, CA, USA).

### 4.4. Sample Collection and Processing

One hour after the last administration, animals were anesthetized using 1% isoflurane (Esteve, Barcelona, Spain) and euthanized by decapitation. This time period was based on the pharmacokinetic knowledge of scFv-h3D6 in this mouse model [25]. Brains were immediately removed from the skulls, rinsed in cold PBS, weighed, and dissected on ice. One hemisphere was rapidly fixed by immersion in 4% paraformaldehyde (PFA) diluted in PBS for 48 h. Samples were embedded in paraffin following common procedures, serially sectioned in the coronal plane (10-µm thick), and mounted on SuperfrostTM Plus microscope slides (Thermo Fisher Scientific, Waltham, MA, USA). Dissected areas (cerebral cortex and hippocampus) of the other hemisphere were mechanically homogenized using a tissue homogenizer (Sigma-Aldrich Química SL, Saint Louis, MO, USA) in cold TBS buffer (pH 7.6) supplemented with a protease inhibitor cocktail (Roche). After a brief sonication step (1 cycle of 35 s, at 35% duty cycle, and output 4, in a Dynatech Sonic Dismembrator ARTEK 300 (Biologics Inc., Manassas, VA, USA), the samples were centrifuged at 100,000× *g* at 4 °C for 1 h. The supernatant was labeled as the TBS-soluble fraction. The pellet was re-suspended in cold TBS, supplemented with an inhibitor cocktail and 2% SDS. Sonication and centrifugation steps were repeated, and the supernatant was labeled as the SDS-soluble fraction. Finally, the pellet was re-suspended in 70% formic acid and, after sonication and centrifugation, the supernatant was recovered and dried overnight in a vacuum concentrator (Savant SpeedVac, Thermo Fischer Scientific). The dried formic extract was re-suspended in DMSO and labeled as the FA-soluble fraction. All fractions were immediately stored at −80 °C until further use.

### 4.5. Immunohistochemistry

Sections were deparaffinized by immersion in xylene and rehydrated in serial dilutions of alcohol. Endogenous peroxidase activity was blocked by immersion in 3% H_2_O_2_ in pure methanol for 10 min. Antigen retrieval was achieved by immersion in 0.01M citrate buffer (pH 6.0) supplemented with 0.1% Tween-20 (Sigma-Aldrich Química SL) at 96 °C for 20 min. Then, tissues were cooled at RT in PBS-T (PBS (pH 7.6), 0.1% Tween-20) and incubated in blocking buffer (5% normal goat serum (NGS, Sigma-Aldrich Química SL) and 5% bovine serum albumin (BSA, Sigma-Aldrich Química SL) in PBS-T) to avoid non-specific binding. Slides were incubated overnight at 4 °C with the corresponding primary antibody (mouse anti-Aβ 6E10 mAb, 1:200, Covance Signet, ref. SIG-39320-200, lot. D13BF00601; rabbit polyclonal anti-glial fibrillary acidic protein, GFAP, 1:200, DAKO, ref. Z033401-2, lot. 20010594; rabbit polyclonal anti-ionized calcium binding adaptor molecule 1, Iba1, 1:200, Wako Pure Chemical Industries, ref. 019-19741, lot. WDR2342; mouse monoclonal anti-human tau mAb, HT7, 1:100 Thermo Scientific, ref. MN1000, lot. TA2494341; and AT8, 1:50, mouse monoclonal anti-human phospho-tau (Ser202, Thr205), Thermo Scientific, ref. MN1020, lot. TF2573772). After incubation with the corresponding secondary antibody (Mouse ExtrAvidin Peroxidase Staining Kit antibody, Sigma-Aldrich Química SL, ref. EXTRA2-1KT; anti-rabbit IgG BA-1000 and streptavidin, Vector Laboratories, ref. BA-1000), the slides were visualized using 3-3′ Diaminobenzidine (DAB, Sigma-Aldrich Química SL).

### 4.6. Immunofluorescence

Sections were deparaffinized and rehydrated, as described in Section 2.5. After antigen retrieval and blocking, samples were incubated at 4 °C overnight with the corresponding primary antibody (6E10, 1:200; GFAP, 1:200; Iba1, 1:200; HT7, 1:100; and rabbit anti-scFv-h3D6, 1:100 [22]. Sections were extensively washed and incubated at RT for 1 h with the corresponding secondary antibody (goat anti-mouse IgG, Cy3 conjugated, Chemicon, Millipore, ref. AP124C; and goat anti-rabbit IgG, Alexa488 conjugated, Chemicon, Millipore, ref. AP123F). Finally, samples were cover-slipped with antifade Vectashield mounting medium (Vector Laboratories, Burlingame, CA, USA).

### 4.7. Image Capture and Processing

Bright-field images were captured using a Leica DMRB microscope and a Leica DFC 500 camera with a Leica PL Fluotar lens. Confocal imaging was performed with a spectral Fluoview-1000 microscope (Olympus, Shinjuku, Japan) by capturing 10 optical sections with the UPLSAO0 objective lens. The cerebral sections corresponded to the range of coordinates from Figure 43 (interaural 2.34 mm and Bregma −1.46 mm) to Figure 48 (interaural 1.74 mm and Bregma −2.06 mm) in *The Mouse Brain in Stereotaxic Coordinates* [75].

The number of plaques and percentage of 6E10-immunoreactive area were semi-quantified using the ImageJ software (NIH, Bethesda, MD, USA).

### 4.8. Western Blotting

Tau, GFAP, Iba1, and LRP1 protein levels in the SDS-soluble fraction of the hippocampus were semi-quantified by immunoblotting analysis. After blocking, membranes were incubated at 4 °C overnight with the corresponding primary antibody (HT7, 1:100, mouse monoclonal anti-human tau mAb; AT8, 1:50, mouse monoclonal anti-human phospho-tau (Ser202, Thr205); anti-GFAP, 1:2500; Iba1, 1:500; mouse monoclonal anti-LRP1 1:1000; abcam, ref. ab28320; and rabbit monoclonal anti-glyceraldehyde 3-phosphate dehydrogenase, GAPDH, 1:20000, abcam, ref. ab9483). Membranes were incubated at RT for 1 h with the corresponding secondary antibody (goat peroxidase conjugated anti-rabbit, 1:2000, Bio-Rad, ref. 1706515; and goat peroxidase conjugated anti-mouse, 1:2000, Sigma-Aldrich Química SL, ref. A3673. Finally, Western blotting (WB) results were visualized using a lumiol/peroxidase solution 1:1 (Clarity ECL Western Blot Substrate kit, Bio-Rad, ref. 170-5060). ImageJ was used for densitometric analyses, and bands were normalized to GAPDH loading controls for each sample.

### 4.9. Enzyme-Linked Immunosorbent Assays (ELISAs)

Aβ_40_, Aβ_42_, TNFα, IL-6, IL-1β, and IL-33 levels were quantified by ELISA following the protocol recommended by the manufacturer (Human Aβ_40_ ELISA kit, Invitrogen, ref. KHB3481; Human Aβ_42_ ELISA kit, Invitrogen, ref. KHB3441; Mouse Duoset TNFα ELISA, RD Systems, ref.DY410-05; Mouse Duoset IL-6 ELISA, RD Systems, ref. DY401-05; Mouse Duoset IL-1β ELISA, RD Systems, ref. DY406-05; Mouse Duoset IL-33 ELISA, RD Systems, ref. DY3626-05). The data were normalized to the total amount of protein in each sample (Pierce BCA Protein Assay Kit, Thermo Scientific, ref. 23225).

## Figures and Tables

**Figure 1 ijms-21-06630-f001:**
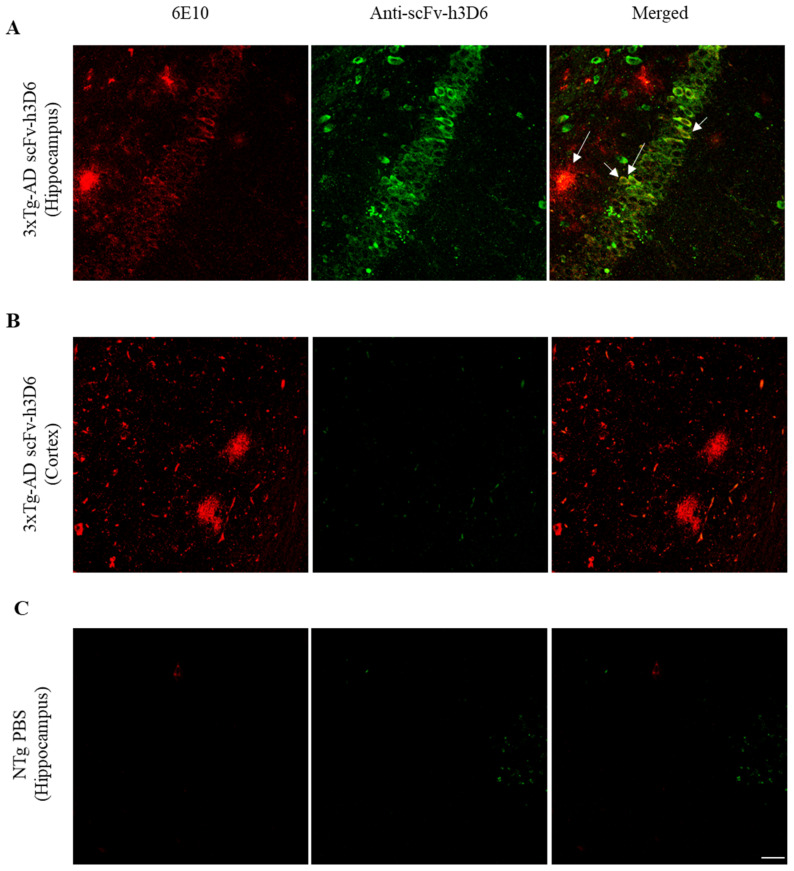
ScFv-h3D6 is mainly distributed throughout the hippocampus and co-localizes with extra and intracellular Aβ. Aβ in red (left), scFv-h3D6 in green (center), and Aβ/scFv-h3D6 merged (right). Arrows indicate scFv-h3D6-Aβ colocalization. (**A**) Labeling in 3xTg-AD hippocampus. (**B**) Labeling in 3xTg-AD cortex. (**C**) Labeling in the hippocampus of an untreated non-transgenic (NTg) mouse to determine the background noise. Scale bar, 25 µm.

**Figure 2 ijms-21-06630-f002:**
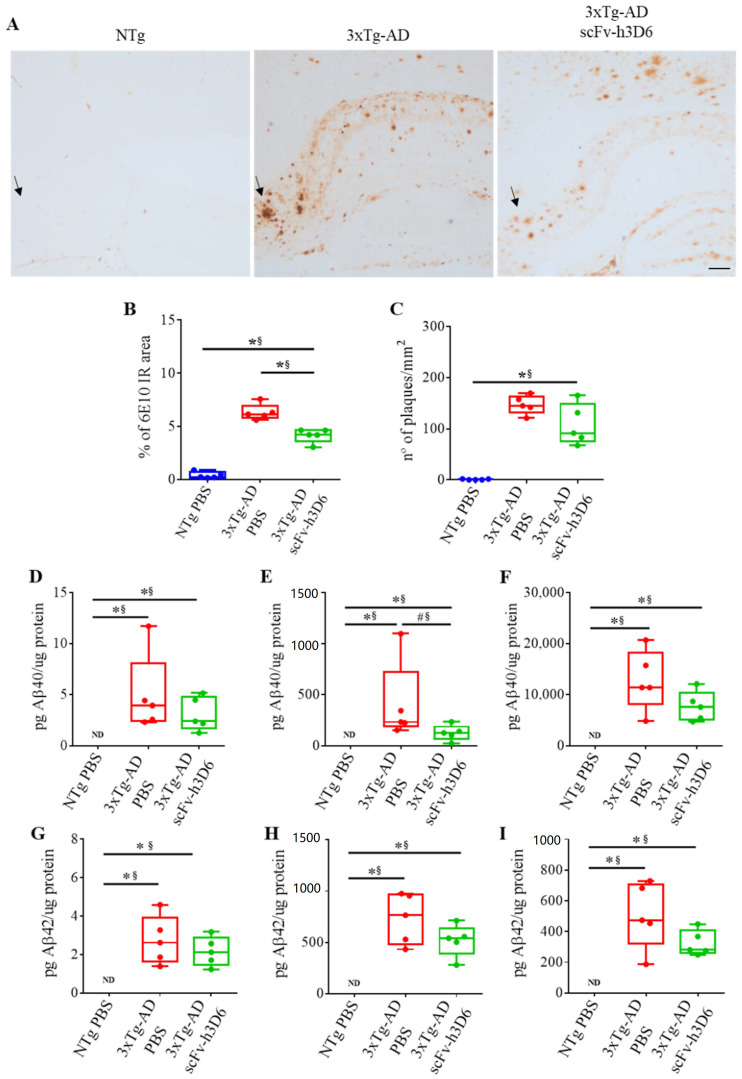
ScFv-h3D6 reduces Aβ burden in the hippocampus. Arrows indicate Aβ accumulation in the subiculum. (**A**) Aβ labeling in untreated NTg (left), untreated 3xTg-AD (center), and scFv-h3D6-treated 3xTg-AD (right) mice. Aβ is mainly accumulated in the subiculum of 3xTg-AD animals and is reduced after scFv-h3D6 treatment. Scale bar, 100 µm. (**B**) Percentage of 6E10-immunoreactive area. (**C**) The number of senile plaques per mm^2^. (**D**–**F**) Aβ_40_ levels as quantified by ELISA (in picograms) and normalized by the total amount of protein as quantified by the BCA assay (in micrograms). (**D**) Aβ_40_ levels in the TBS-soluble fraction. (**E**) Aβ_40_ levels in the SDS-soluble fraction. (**F**) Aβ_40_ levels in the FA-soluble fraction. (**G**–**I**) Aβ_42_ levels as quantified by ELISA (in picograms) and normalized by the total amount of protein as quantified by the BCA assay (in micrograms). (**G**) Aβ_42_ levels in the TBS-soluble fraction. (**H**) Aβ_42_ levels in the SDS-soluble fraction. (**I**) Aβ_42_ levels in the FA-soluble fraction. Three replicates for each experiment. Data are expressed as medians in box plots, and whiskers represent the minimum and maximum values. Statistical differences were assessed with the non-parametrical Mann-Whitney *U*-test. * Significant differences *p* ≤ 0.05; ^#^ Marginal significance *p* ≤ 0.1; § r ≥ 0.68.

**Figure 3 ijms-21-06630-f003:**
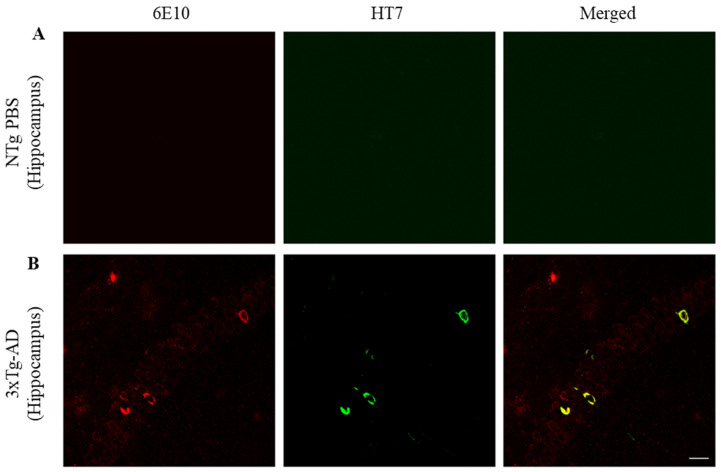
NTFs co-localize within Aβ-containing neurons in CA1 of the hippocampus. (**A**) On the left Aβ in red (6E10), in the center tau in green (HT7), and on the right Aβ/tau merged. Labeling in the hippocampus of an untreated NTg mouse to determine the background noise. (**B**) Labeling in 3xTg-AD hippocampus. Scale bar, 25 µm.

**Figure 4 ijms-21-06630-f004:**
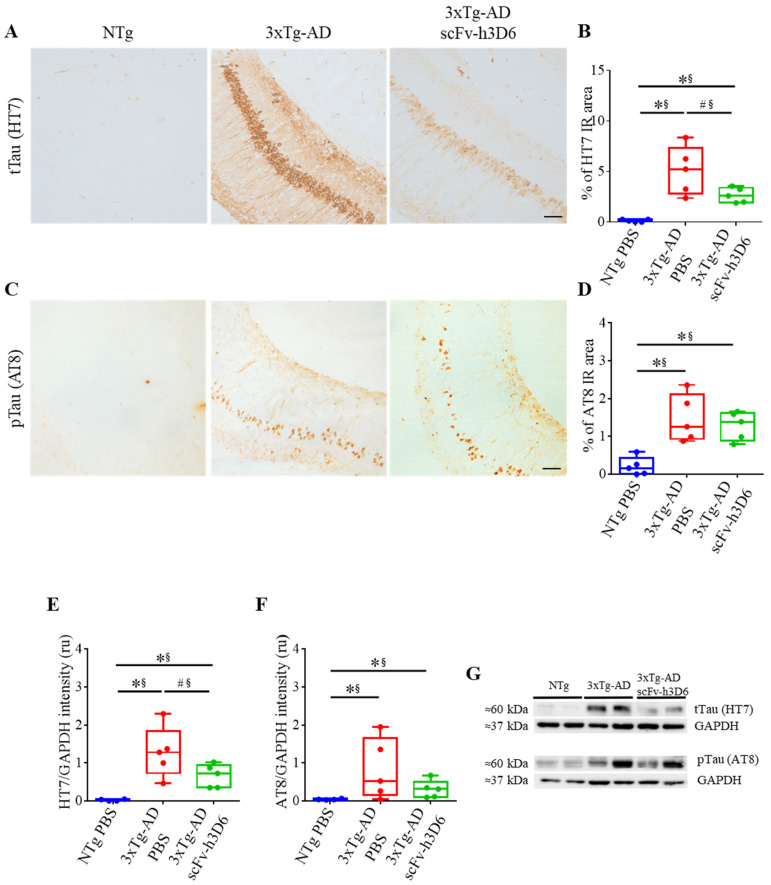
ScFv-h3D6 treatment tends to decrease the total tau load in the hippocampus. (**A**) Total tau (HT7) labeling in untreated NTg (left), untreated 3xTg-AD (center), and scFv-h3D6-treated 3xTg-AD (right) mice. (**B**) Percentage of HT7-immunoreactive area. (**C**) Ser202 and Thr205 Phosphorylated tau levels (AT8) labeling in untreated NTg (left), untreated 3xTg-AD (center), and scFv-h3D6-treated 3xTg-AD (right) mice. Scale bar 25 µm. (**D**) Percentage of the AT8-immunoreactive area. (**E**) Total tau levels (HT7) in the SDS-soluble fractions, as measured by Western blotting and normalized by GAPDH levels. Intensity is expressed in relative units. (**F**) Ser202 and Thr205 Phosphorylated tau levels (AT8) in the SDS-soluble fractions, as measured by Western-blotting and normalized by GAPDH levels. Intensity is expressed in relative units. (**G**) Representative Western-blot image of the total and phosphorylated tau protein in the SDS-soluble fractions from the untreated NTg, untreated 3xTg-AD, and scFv-h3D6-treated 3xTg-AD mice. Three replicates for each experiment. Data are expressed as medians in box plots, and whiskers represent the minimum and maximum values. Statistical differences were assessed with the non-parametrical test *U* Mann-Whitney. * Significant differences *p* ≤ 0.05; ^#^ Marginal significance *p* ≤ 0.1; § r ≥ 0.68.

**Figure 5 ijms-21-06630-f005:**
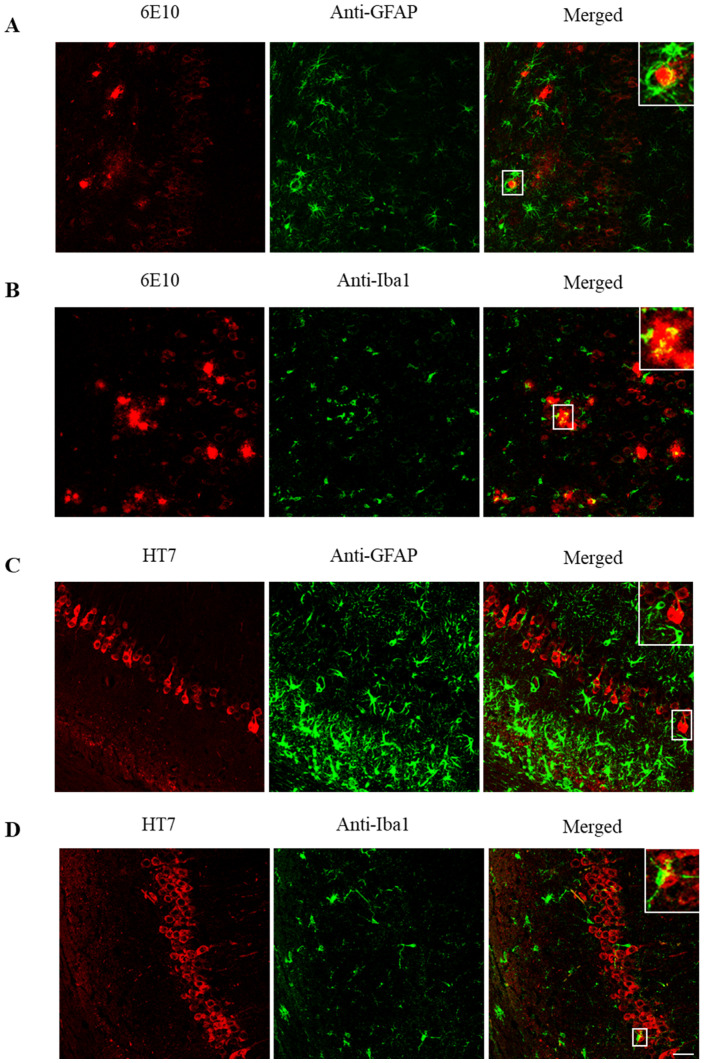
Astroglia and microglia surround Aβ-peptide and tau protein in the hippocampus. (**A**) Aβ and GFAP (astroglia) labeling in 3xTg-AD mice. Aβ in red (left), glial fibrillary acidic protein (GFAP) in green (center), and Aβ/GFAP merged (right). Astrocytes surround senile plaques. (**B**) Aβ and Iba1 (microglia) labeling in 3xTg-AD mice. Aβ in red (left), Iba1 in green (center), and Aβ/Iba1 merged (right). Microglia surrounds senile plaques. (**C**) Tau and GFAP labeling in 3xTg-AD mice. Tau in red (left), GFAP in green (center), and tau/GFAP merged (right). Astrocytes surround tau deposits. (**D**) Tau and Iba1 labeling in 3xTg-AD mice. Tau in red (left), Iba1 in green (center), and tau/GFAP merged (right). Microglia surrounds tau deposits. Scale bar, 25 µm.

**Figure 6 ijms-21-06630-f006:**
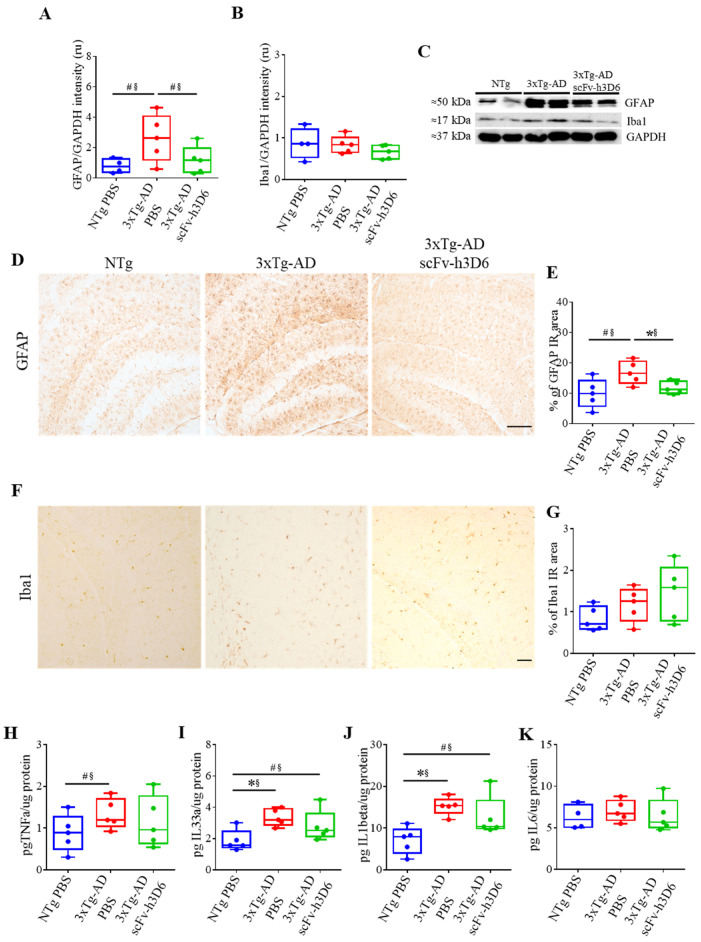
ScFv-h3D6 does not elicit any detectable neuroinflammatory response in the hippocampus. (**A**) GFAP levels are measured by Western blotting and normalized by GAPDH levels. Intensity is expressed in relative units. (**B**) Iba1 levels measured by Western blotting and normalized by GAPDH levels. Intensity is expressed in relative units. (**C**) Representative Western blot image of GFAP and Iba1 in NTg, 3xTg-AD PBS, and 3xTg-AD scFv-h3D6 treated mice. (**D**) GFAP labeling in untreated NTg (left), untreated 3xTg-AD (center), and scFv-h3D6-treated 3xTg-AD (right) mice. Scale bar, 100 µm. (**E**) Percentage of GFAP-immunoreactive area. (**F**) Iba1 labeling in untreated NTg (left), untreated 3xTg-AD PBS (center), and scFv-h3D6-treated 3xTg-AD (right) mice. Scale bar, 25 µm. (**G**) Percentage of Iba1-immunoreactive area. (**H**–**K**) Interleukin levels as measured by ELISA (in picograms) and normalized by the total amount of protein as quantified by the BCA assay (in micrograms) in the TBS-soluble fraction. (**F**) TNF-α levels. (**G**) IL-33 levels. (**H**) IL-1β levels. (**I**) IL-6 levels. Three replicates for each experiment. Data are expressed as medians in box plots, and whiskers represent the minimum and maximum values. Statistical differences were assessed with the non-parametrical test *U* Mann-Whitney. * Significant differences *p* ≤ 0.05; ^#^ Marginal significance *p* ≤ 0.1; § r ≥ 0.68.

**Figure 7 ijms-21-06630-f007:**
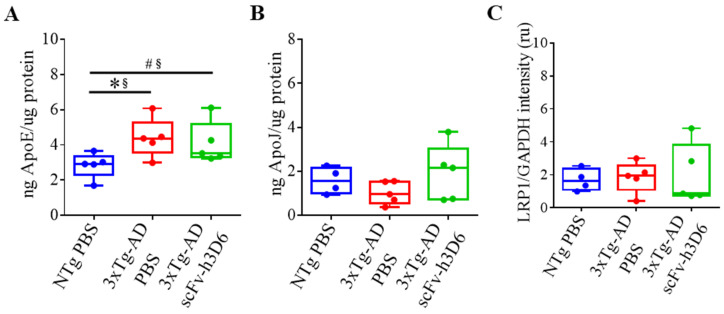
Apolipoprotein levels in the hippocampus are not modified after scFv-h3D6 treatment. (**A**,**B**) ApoE and apoJ levels as measured by ELISA (in nanograms) and normalized by the total amount of protein as quantified by the BCA assay (in micrograms) in the TBS-soluble fraction. (**A**) ApoE levels. (**B**) ApoJ levels. (**C**) LRP1 levels were measured by Western blotting and normalized by GAPDH levels. Intensity is expressed in relative units. Three replicates for each experiment. Data are expressed as medians in box plots, and whiskers represent the minimum and maximum values. Statistical differences were assessed with the non-parametrical test *U* Mann-Whitney. * Significant differences *p* ≤ 0.05; ^#^ Marginal significance *p* ≤ 0.1; § r ≥ 0.68.

**Table 1 ijms-21-06630-t001:** Comparison between the *p*-values (Mann-Whitney U-test) and the r-coefficients (using the obtained *U*-values in the Wendt formula [74]). Considering the size of the groups in this study (*n* = 5), r-coefficients ≥0.68 are significant at *p*-value ≤ 0.05, whereas values ≥0.92 (more restrictive because the r-coefficient ranges from 0 to 1) are significant at *p*-value ≤ 0.05 to *p* ≤ 0.01 [74].

	NTg vs. 3xTg-AD		3xTg-AD (−) vs. 3xTg-AD (+)	
	*p*-Value	r-Coeff.	*p*-Value	r-Coeff.
% of 6e10 IR area (Figure 2B)	0.0079 **	1 **	0.0079 **	1 **
n° of plaques/mm^2^ (Figure 2C)	0.0079 **	1 **	0.1508	0.60
TBS-fraction pg Aβ_40_/µg protein (Figure 2D)	0.0079 **	1 **	0.5317	0.28
SDS-fraction pg Aβ_40_/µg protein (Figure 2E)	0.0079 **	1 **	0.0556 ^#^	0.76 *
FA-fraction pg Aβ_40_/µg protein (Figure 2F)	0.0079 **	1 **	0.2222	0.52
TBS-fraction pg Aβ_42_/µg protein (Figure 2G)	0.0079 **	1 **	0.4127	0.36
SDS-fraction pg Aβ_42_/µg protein (Figure 2H)	0.0079 **	1 **	0.3095	0.44
FA-fraction pg Aβ_42_/µg protein (Figure 2I)	0.0079 **	1 **	0.1508	0.60
% of HT7 IR area (Figure 4B)	0.0079 **	1 **	0.0952 ^#^	0.68*
% of AT8 IR area (Figure 4D)	0.0079 **	1 **	0.8016	0.12
HT7/GAPDH intensity (U.A.) (Figure 4E)	0.0079 **	1 **	0.0952 ^#^	0.68 *
AT8/GAPDH intensity (U.A.) (Figure 4F)	0.0159 *	0.92 **	0.5317	0.28
GFAP/GAPDH intensity (U.A.) (Figure 6A)	0.0556 ^#^	0.76 *	0.0952 ^#^	0.68 *
Iba1/GAPDH intensity (U.A.) (Figure 6B)	0.7857	0.12	0.1508	0.60
% of GFAP IR area (Figure 6E)	0.0556 ^#^	0.76 *	0.0317 *	0.84 *
% of Iba1 IR area (Figure 6G)	0.2222	0.52	0.5317	0.28
pg TNF_α_/µg protein (Figure 6H)	0.0952 ^#^	0.68 *	0.5317	0.28
pg IL33_α_/µg protein (Figure 6i)	0.0159 *	0.92 **	0.2222	0.52
pg IL1_β_/µg protein (Figure 6J)	0.0079 **	1 **	0.1508	0.60
pg IL6/µg protein (Figure 6K)	0.6667	0.20	0.5317	0.28
ng ApoE/µg protein (Figure 7A)	0.0317 *	0.84 *	0.6667	0.20
ng ApoJ/µg protein (Figure 7B)	0.3889	0.36	0.2222	0.52
LRP1/GAPDH intensity (U.A.) (Figure 7C)	0.5159	0.28	0.9444	0.04

** Significant differences *p* ≤ 0.01; * Significant differences *p* ≤ 0.05; ^#^ Marginal significance *p* ≤ 0.1. 3xTg-AD (−), untreated. 3xTg-AD (+), scFv-h3D6-treated.

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
