# Peer review of "Both Amyloid-β Peptide and Tau Protein Are Affected by an Anti-Amyloid-β Antibody Fragment in Elderly 3xTg-AD Mice"

_ijms, 2020, doi:10.3390/ijms21186630_

Round 1

Reviewer 1 Report

The manuscript from Roda A.R. et al., describes the efficacy of scFv-h3D6 in 22 month-old 3xTgAD animals. The authors report data concerning the investigation of Ab and tau staining by employing immunohistochemistry, immunofluorescence and biochemical approaches on different area of animal brains. Although the manuscript is well-written, it suffers from major technical issues and some of the claims are not always supported by the presented experimental data.

Major points:

Figure 1A: immunohistostaining of brain sections from treated 3xTgAD should show that ScFv-h3D6 is mainly distributed throughout the hippocampus and co-localizes with extra and intracellular Aβ.

  1. Here, Aß is detected by using the 6E10 antibody, which should recognize the 1-16 aa sequence in the peptide. This epitope is retained when APP is processed by a-secretase to produce the soluble aAPPs (that can be recognized by 6E10, too). The authors should describe how they can claim that the 6E10 is recognizing Aβ (intended as “free peptide”).
  2. Colocalization analysis between Aß and ScFV-h3D6 should be supported by statistical elaboration: how many sections have been analysed? Which colocalization parameter (Pearson, Mander’s coefficient??) has been employed?
  3. To investigate extra- and intra-cellular Aβ the labeling of cell nuclei is mandatory.

Figure 3: This figure is confusing to me. Panel A): This staining should show colocalization between NTFs and Ab containing neurons in the CA1 of the hippocampus. Unfortunately, contrary to Figure 1A, where Aß was present, here the Ab is not present…maybe the panels A and B should be inverted…maybe! Again, as in figure 1, colocalization must be supported by statistical analysis (Mander’s or Pearson….coeff).

Figure 4: After the treatment with scFv-h3D6, a tendency for a reduction in the total tau levels was found and non-significant differences in phosphorylated tau levels, as quantified with the mAb AT8, were observed after the treatment. The authors report at line 297 : “It should be noted that a large range in tau levels is inherent to the 3xTg-AD model, for both total and phosphorylated tau, and particularly for the latter. However, it is interesting that scFv-h3D6 treatment reduced this dispersion”. This statement should be better discussed in the light of the novelty of treatment with the new antibody.

Figure 5: the interplay between Aß, tau and glial cells is a crucial issue in AD pathogenesis. The image in the panel A is not sufficient to report astroglia surrounding Aß peptide. The area reported in the ROI is too small to claim that interplay.

Lines 90-91: …” the mechanism underlying the Aβ-peptide withdrawal is engulfment of the scFv-h3D6/Aβ complex, by the glial cells…”. I do not see any engulfment of scFv by these cells, at least in the images shown.

Minor points:

Line 300, Figure 4: it is reported in the text twice.

Discussion:

starting from line 495: Please improve discussion on APOE mechanism by reporting the following reference: “New Insights into the Molecular Bases of Familial Alzheimer's Disease. J Pers Med. 2020 Apr 19;10(2):E26. doi: 10.3390/jpm10020026.”

lines 424-427: When discuss the “use of bispecific antibodies combining an anti-Aβ mAb with an anti-transferrin R antibody that has been used as a BBB transporter, could improve the distribution of therapeutic mAbs in the brain, spreading its activity and increasing the efficacy of treatment”. Please report the following ref.:  “APP Maturation and Intracellular Localization Are Controlled by a Specific Inhibitor of 37/67 kDa Laminin-1 Receptor in Neuronal Cells. Bhattacharya A, et al. Int J Mol Sci. 2020 Mar 4;21(5):1738. doi: 10.3390/ijms21051738.” Here, the authors report that APP is able to be relocated in the transferrin-positive recycling endosomes, which could be the major intracellular site for Ab production.

Line 448: when reporting the interplay between Ab and tau is well documented, please insert this reference: “Attempt to Untangle the Prion-Like Misfolding Mechanism for Neurodegenerative Diseases. Int J Mol Sci. 2018 Oct 9;19(10):3081. doi: 10.3390/ijms19103081.”

Reviewer 2 Report

Manuscript # ijms-892090

Both Amyloid-β and Tau-like Pathologies are Ameliorated by an Anti-amyloid-β Antibody Fragment in Elderly 3xTg-AD Mice

Comments of the reviewer

In the aim to improve the manuscript, several changes should be included in the final version of the manuscript.

INTRODUCTION

  1. The authors begin by explaining the clinical signs and symptoms of Alzheimer's disease. In this clinical context and as a complement to that described by the authors, it is suggested that they provide information on the incidence and / or prevalence of this disease in Europe, indicating whether it is more or less prevalent in men and women, and the ages at which the signs and symptoms of the disease are most clinically evident. Likewise, that it provides information on the social and health costs of this disease. All this complements the clinical information provided by the authors and gives a more precise vision of the problems of this disease. Please include all this new information in the final version of the manuscript.

MATERIALS AND METHODS

  1. The authors make a correct justification of why they only use females in the study. Epidemiological evidence of Alzheimer's disease indicates that it is a more prevalent disease in women than in men, most likely because the life expectancy of women is higher than that of men in Western societies, and especially in Europe. Therefore, it is very correct that the authors have carried out their research on females. This information should be included in the corresponding section of the methodology.
  2. It would be very convenient for the authors to include in the corresponding "animals" section technical information on the strain of mice they have used, which corresponds to the triple-transgenic strain for Alzheimer's disease. The minimum information that they should include would be: estimated life expectancy, that is, the maximum age at which both males and females live, and signs and symptoms that these animals present. This minimum information, which can be obtained from the animal data sheet, will make the manuscript more robust, especially for less expert readers, where they will be able to verify that the animal model used has the most similarities to human Alzheimer's disease, and also that the authors have used very advanced ages of the disease, similar to what happens in humans. Please include this information in the final version of the manuscript.
  3. How many histological sections per animal and experimental group have been analyzed? Please include this information in the final version of the manuscript.

RESULTS

  1. In the feet of Figures 4A, 4B, 6D, 6E it is indicated that there is scale bar = 25 µm, however in the histological images no scale bar is observed, perhaps it is white. Please change the color of the scale bar or add it.

DISCUSSION

  1. The results obtained by the authors are very clear, however, they do not carry out a mechanistic discussion to understand why the administration of the antibodies in old animals with Alzheimer's reduces beta-amyloid plaques and the tau protein, or why it reduces gliosis. What does this treatment do that causes neurons to not synthesize and secrete the typical proteins of Alzheimer's disease? How does this treatment reduce cerebral gliosis? Can this reduction in gliosis influence a lower level of beta-amyloid and/or tau protein? Please clarify these points and include this information in the final version of the manuscript.
  2. In none of the paragraphs of the discussion the authors do indicate whether the results derived from the present study are new or if they are the first time they have been obtained. Considering that, at present, there are no efficient therapies for Alzheimer's disease, the results of the present study are relevant, and this relevance should be reflected in the discussion of the manuscript. Please include this information in the final version of the manuscript.

Reviewer 3 Report

Alzheimer’s disease (AD) is the most common dementia worldwide that, according to the amyloidogenic hypothesis, may arise from the early accumulation of the amyloid-β (Aβ) peptide and its subsequent induction of Tau protein hyperphosphorylation, synaptic dysfunction, neuronal death and, ultimately the cognitive/behavioral/psychological impairments. Since AD remains incurable, the Authors aimed to evaluate the efficacy of the peripheral administration of scFv-h3D6 (a single-chain variable fragment that was able to mitigate the amyloid burden in the adult 3xTg-AD mice) in older female mice with evident AD neuropathological hallmarks. The Authors claim that treatment with scFv-h3D6 reduced the hippocampal Aβ staining and also tended to decrease Tau protein levels in elderly 3xTg-AD female mice without affecting their apolipoproteins E or J or inflammatory state.

Conceptually, this subject could be of high relevance in terms of AD therapeutics. Although the study was well-designed and the Authors performed numerous determinations and complementary evaluations in the 3xTg-AD female mouse model, I found some writing issues that must be taken in account to improve the quality of this manuscript:

  1. How much protein was loaded into the Western blotting gels and how did Authors quantify such protein levels?
  2. When presenting the typical images from immunohistochemistry, please provide lower magnification images containing the whole area from which they were taken. This is particularly relevant in the case of cortical-derived pictures. It would be interesting if the Authors could also complement this data with the corresponding graphs. How many animals and slides/animal were visualized?
  3. The text is hardly readable in some figures/graphs.
  4. In the results from Fig.3, the Authors refer the intracellular localization of neurofibrillary tangles, and the intra- and extracelular deposition of Aβ in the hippocampus. However, from the images provided, it seems that such accumulations were more evident in the non-transgenic mice. Please, comment on this apparent contradiction. In order to assure an easier observation that the accumulations were intra or extracellular, Authors should have included a nuclear or plasma membrane marker to limit the cellular environment.
  5. The legend of Fig. 4 appears repeated in the manuscript.
  6. Please, write the y-axes legends of the graphs in a more accurate manner.
  7. In Discussion, the Authors refer that scFv-h3D6 was mainly found within the hippocampus and its accumulation within the cortex was negligible. However, since the later area has been increasingly described to be also affected in AD, it would be interesting if the Authors evaluated if scFv-h3D6 could exert some indirect cortical effects in elderly 3xTg-AD female mice.
  8. In face of the mounting evidence suggesting that other non-amyloidogenic or -taugenic mechanisms may underlie AD pathophysiology, please comment on the potential pros and cons of using scFv-h3D6.

Round 2

Reviewer 1 Report

the authors answered all my questions and clarified my doubts

Author Response

Thank you